# Neuro-Oncology Multidisciplinary Tumor Board: The Point of View of the Neuroradiologist

**DOI:** 10.3390/jpm12020135

**Published:** 2022-01-20

**Authors:** Simona Gaudino, Carolina Giordano, Francesca Magnani, Simone Cottonaro, Amato Infante, Giovanni Sabatino, Giuseppe La Rocca, Giuseppe Maria Della Pepa, Quintino Giorgio D’Alessandris, Roberto Pallini, Alessandro Olivi, Mario Balducci, Silvia Chiesa, Marco Gessi, Pamela Guadalupi, Rosellina Russo, Chiara Schiarelli, Luca Ausili Cefaro, Giuseppe Maria Di Lella, Cesare Colosimo

**Affiliations:** 1Department of Diagnostic Imaging, Oncological Radiotherapy, and Hematology, UOC Neuroradiology, Fondazione Policlinico Universitario Agostino Gemelli IRCCS and Università Cattolica del Sacro Cuore, 00168 Rome, Italy; carolinagiordano91@gmail.com (C.G.); francesca.magnani193@gmail.com (F.M.); rosellina.russo@policlinicogemelli.it (R.R.); chiara.schiarelli@policlinicogemelli.it (C.S.); luca.ausilicefaro@policlinicogemelli.it (L.A.C.); giudilella@hotmail.com (G.M.D.L.); cesare.colosimo@policlinicogemelli.it (C.C.); 2Department of Diagnostic, Interventional Radiology, Neuroradiology, Garibaldi Hospital, 95122 Catania, Italy; cottonaro.simone90@gmail.com; 3Department of Diagnostic Imaging, Oncological Radiotherapy, and Hematology, UOC Diagnostic Imaging, COVID Center 2, Fondazione Policlinico Universitario Agostino Gemelli IRCCS, 00168 Rome, Italy; amatoinfante@gmail.com; 4Department of Neurosurgery, Fondazione Policlinico Universitario Agostino Gemelli IRCCS and Università Cattolica del Sacro Cuore, 00168 Rome, Italy; giovanni.sabatino@policlinicogemelli.it (G.S.); giularocca86@gmail.com (G.L.R.); giuseppemaria.dellapepa@policlinicogemelli.it (G.M.D.P.); quintinogiorgio.dalessandris@policlinicogemelli.it (Q.G.D.); roberto.pallini@policlinicogemelli.it (R.P.); alessandro.olivi@policlinicogemelli.it (A.O.); 5UOC Neurosurgery, Mater Olbia Hospital, 07026 Olbia, Italy; 6Department of Diagnostic Imaging, Oncological Radiotherapy, and Hematology, UOC Oncological Radiotherapy, Fondazione Policlinico Universitario Agostino Gemelli IRCCS and Università Cattolica del Sacro Cuore, 00168 Rome, Italy; mario.balducci@policlinicogemelli.it (M.B.); silvia.chiesa@policlinicogemelli.it (S.C.); 7UOS Neuropathology, UOC Pathology Fondazione Policlinico Universitario Agostino Gemelli IRCCS, 00168 Rome, Italy; marco.gessi@policlinicogemelli.it; 8Neuroradiology Unit, S. Maria Hospital, 05100 Terni, Italy; pamelaguadalupi@gmail.com

**Keywords:** multidisciplinary tumor board, neuro-oncology, neuroradiology, brain tumors, gliomas, brain metastases, MRI

## Abstract

Background: The multi-disciplinary tumor board (MTB) is essential to quality cancer care and currently recommended to offer the best personalized clinical approach, but little has been published regarding MTBs in neuro-oncology (nMTBs). The aim of the present paper is to describe our nMTB, to evaluate its impact on clinical management decisions, and to assess the role of neuroradiologists. Methods: The retrospective evaluation of the cases discussed at our nMTB from March 2017 to March 2020. From the electronic records, we extracted epidemiological, clinical and other specific data of nMTB. From the radiological records, we calculated data relating to the number, time for revision, and other specifications of MRI re-evaluation. Statistical analysis was performed. Results: a total of 447 discussions were analyzed, representing 342 patients. The requests for case evaluations came from radiation oncologists (58.8%) and neurosurgeons (40.5%), and were mainly addressed to the neuroradiologist (73.8%). The most frequent questions were about the treatment’s changes (64.4%). The change in patient treatment was reported in 40.5% of cases, 76.8% of these were based on the neuroradiologic assessment. A total of 1514 MRI examinations were re-evaluated, employing approximately 67 h overall. The median of the MRI exams reviewed per patient was 3 (min–max 1–12). Conclusions: Our study supported that the multidisciplinary approach to patient care can be particularly effective in managing brain tumors. A review by an expert neuroradiologist impacts patient management in the context of nMTBs, but has costs in terms of the time and effort spent preparing for it.

## 1. Introduction

Multi-disciplinary tumor board meetings (MTBs) are considered essential for quality cancer care and are currently recommended to offer the best personalized clinical approach. Quality cancer care is complex and depends on the careful coordination between multiple treatments and providers, technical information exchange, and regular communication flow between all involved parties [1,2]. MTBs represent a moment of multidisciplinary management of the diagnostic–therapeutic path of cancer patients based on the collegial discussion of individual cases. This is especially helpful in complex cases with a clinically and/or radiologically difficult context in both the diagnosis and follow-up phase.

Hospital tumor boards were first defined by Gross in 1987 as multidisciplinary groups of physicians that meet on a regular basis, to review cancer cases with the primary goal of improving the care of their community’s cancer patients through the exchange of information among participating physicians. Furthermore, the use of MTB consultations can ensure that the cancer patient has access to the best current thinking about cancer management and provide a mechanism for reviewing the quality of professional care [3]. Since then, the number of MTBs has increased exponentially, especially in the last 10 years, as their usefulness in improving diagnostic accuracy, adherence to clinical practice guidelines and protocols, and improving clinical outcomes has been proven.

In 2014, the Royal College of Radiologists (RCR) published a document highlighting the importance of radiologists in MTBs and outlining the necessary requirements for consultant radiologists and radiology departments to participate in the meetings [4]. The appropriate and efficient staffing of radiologists in the MTB is essential to improve team dynamics and communication, the satisfaction of the multidisciplinary team, and patient care [5]. The specialists involved in multidisciplinary care spend a substantial amount of time and effort preparing for the MTB. Given the time and effort spent preparing for and attending MTBs, their efficiency has been the subject of ongoing controversy [5,6,7].

However, most studies have shown that MTB boards are an important asset for the management and treatment of these patients, leading to a significant improvement in the quality of the medical services offered and, possibly, to higher survival rates [8,9,10,11,12,13,14,15,16,17].

The benefits of MTBs include an education for residents, increased adherence to published guidelines, and clinical trial access [18]. The developments in teleconferencing technology have facilitated the interactions during the COVID-19 pandemic. Currently, in Italy, highly specialized centers have subspecialty, organ-specific MTBs. In this scenario, MTBs dedicated to the CNS that are attended by neurosurgeons, neuro-oncologists, and/or radiotherapists, neuroradiologists, and neuro-anatomopathologists are less common.

However, the MTB is an important part of brain cancer evaluation and management. Given the need for the multimodal treatment of glioma and brain metastasis, the diagnosis and management of these patients should follow multidisciplinary tumor board recommendations throughout the disease course. Compared to other MTBs, there is little in the literature regarding MTBs in neuro-oncology, probably because of the requisites necessary for an MTB in this field: neuroradiologists with a skill in neuro-oncology imaging, oncologic neurosurgical high-volume centers, specialized oncologists (or radiation oncologist), and pathologists [18,19,20].

The neuroradiologist is essential in MTBs, as all cases require the evaluation or re-evaluation of imaging in light of the clinical-anamnestic history and therapeutic perspectives.

In addition to the time dedicated to the meetings, the neuroradiologist is also required to make a variable commitment to preparation for the meetings and the retrospective re-evaluation of the images, the interpretation of which is more reliable by the integration of precise and in-depth anamnestic, clinical, and procedural data and the surgical procedures and therapies performed (for instance, the type of treatment, doses, and timing).

The aim of the present study is to describe the neuro-oncology MTB (nMTB) at our hospital, Fondazione Policlinico Universitario Agostino Gemelli IRCCS (FPG), to evaluate its impact on clinical management decisions and assess the role of the neuroradiologist in terms of the number of cases evaluated, the time taken for the analysis of cases, the correlations between exams acquired on-site and off-site, and its weight in changing patient treatment (COPT).

## 2. Materials and Methods

### 2.1. Clinical Data

At our institution, the nMTB is a weekly meeting, attended by radiation oncologists, oncologists, neurosurgeons, neuropathologists, and neuroradiologists. The case presentations of MTB include the patient’s medical history, clinical presentation, diagnostic studies, and a “pre-conference” treatment plan. Questions are then asked to one or more specialists to decide on the best possible treatment plan for the patient; after the case discussion, if the final decision does not correspond to the one proposed at the beginning, it is considered as a change in patient treatment (CPT). The interventions of the physicians and the final decision concerning each individual patient is formally transcribed and recorded in the MTB report. After obtaining institutional review board approval, we performed a retrospective analysis of all the cases discussed at our nMTB from March 2017 to March 2020. From the electronic medical records, we abstracted the following data: patient age and gender, final diagnosis, proposing physician, question(s) for MTB, specialist to whom the question was addressed, and CPT. From the radiology records of the nMTB, we calculated the number of MRI exams evaluated by the neuroradiologists, how many exams were performed inside and outside our hospital, and the time for the cases’ evaluation.

### 2.2. Statistical Analyses

All data were analyzed with a dedicated software (SPSS for Windows, version 24.0; IBM, Chicago, IL, USA). Continuous variables were evaluated by the Kolmogorov–Smirnov test and Shapiro–Wilk test, which showed the normal distributions of the examined variables. Student’s *t*-test was applied for continuous variables analysis in paired samples, and Pearson/Spearman correlations were used to assess the relationships between continuous variables. One-way ANOVA was performed to compare the intergroup and intragroup variability. Bonferroni’s correction was applied as appropriate. Subgroup analysis was performed as needed. The data were considered statistically significant at *p* < 0.05.

## 3. Results

The average number of cases discussed for the session ranged from 6 to 15 (mean 10.5), with a total of 447 discussions analyzed, representing 342 patients (48.4% female, 51.7% male): 259 were unique patients, 43 were discussed twice, 94 three times, 42 four time, and 9 five times. The median patient age was 58 years (range: 22–86 years). In order of frequency, the tumors types were: glioblastoma (GB) 30.4%, metastasis 27.5%, anaplastic astrocytoma (AA) and meningioma 10.3%, high grade tumor (other than GB and AA) 8.3%, oligodendroglioma (ODG) 6.7%, other low grade tumor 4.5%, and low grade astrocytoma 2% (Figure 1).

The requests for case evaluations came almost exclusively from the radiation oncologists (58.8%) and neurosurgeons (40.5%), and were mainly addressed to the neuroradiologists (73.8% vs. 11.2% neurosurgeon, 7.8% radiation oncologist, 6.5% pathologist, and 0.7% others; Table 1).

Most of the cases were presented by the treating physician to make plans for management. The submitted questions were about the imaging response to non-surgical therapy (64.4%), evaluation of surgical (excision or biopsy) (12.3%) and non-surgical (10.5%) treatment options, imaging diagnosis (11.6%), or histological and/or molecular data (1.1%). Consensus recommendations after nMTB mainly included imaging follow-up with variable timing (according to regular protocol or closer), modality (MRI, PET), and MRI techniques (e.g., higher magnetic field and non-morphological MRI) (40.7%), medical and surgical treatment (55.7%), and molecular study integration (3.6%; Table 2).

The image reinterpretation was discordant with the initial report in 30.9% of the cases, supporting CPT, but this was 40.5% when adding the cases with CPT without radiological discrepancies.

A total of 1514 MRI examinations for 447 patients were evaluated, with a mean number of 3.4 ± SD 2.0 (min–max: 1–12) MRI examinations per patient (Table 3).

Of the 1514 MRI examinations, 701 were non-FPG and belonged to 159 cases brought up for discussion at our nMTB. There was a significant correlation between an increasing number of all MRI examinations per case and the number that were non-FPG (*p* < 0.05). Between the different brain tumor histotypes, ODG was associated with the highest number of MRI examinations reviewed per case, with a maximum of 12 reviewed for a patient (Figure 2).

The time employed to review all MRI exams was 67 h overall, with a mean of 9.0 min for cases (±2.2 SD, min–max 3–21 min). The time required for the neuroradiological evaluation for the nMTB increased linearly with the increase in number of MRI examinations to be reviewed (Figure 3).

We found a significant difference between GB, meningioma, and ODG concerning the evaluation time per case (*p* < 0.05) (Table 4)

We calculated the time needed to re-evaluate the imaging exams in relation to the variable “number of external MRIs”, both for the overall number of exams and for each number of MRIs, from 1–12 exams. The time needed to re-evaluate the MRIs increased with the number of external exams (Figure 4).

## 4. Discussion

MTBs are formal meetings in which networks of specialists devoted to the care of cancer patients meet to discuss diagnosis and management. Even if the evidence on the impact of MTBs on clinical practices is still lacking, the multidisciplinary approach remains the best way to deliver the complex care needed by cancer patients [21]. However, publications regarding MTB in neuro-oncology are limited [18,19,20,22]. The nMTBs involve a core group of neurosurgeons, neuroradiologists, radiation oncologists, and medical oncologists and neuropathologists, as well as other ancillary members of the healthcare team. At our institution, the nMTB started in March 2016, and, since 2017, has taken to meeting on a weekly basis with a more structured organization in terms of the specialists present, presentation of cases, organization of clinical records, collection of imaging exams, and transmission of the results. We discussed primary and secondary brain tumors, whereas neoplastic spine diseases belong to the spinal board meeting and children CNS tumor discussions belong to the pediatric nMTB.

The frequencies of the histotypes discussed at our MTB partly reflects the frequency of these tumors in the population. The histotypes most often brought into discussion were GB and metastases, and partly their management complexity, which deserves multidisciplinary discussion. GB and metastases can lead to greater uncertainty in the MRI diagnosis, require abrupt treatment (often combined and/or repeated), and multiple follow-up MRI examinations. Lower-graded pathologies, with more typical imaging features, and more conservative treatments, such as meningiomas and neuromas, were less frequently brought into question.

As reported by several authors, in both the MTB of neuro-oncology or of others sectors, oncologists presented most of the cases (58.8%) [19,23], whereas surgeons (in our case neurosurgeons) have proved to be much more active than in other MTBs, presenting 40.5% of the cases. This strong and active participation of neurosurgeons is because the nMTB project was proposed by our neurosurgeons, who also maintain a very high number of participants in each session.

Through MTB reports, we traced the final treatment decision and whether or not it corresponded to the a “pre-MTB” treatment plan. In a total of 76.8% CPT, 40.5% of it was based on neuroradiologic assessment. A practical example is therefore reported: a radiation oncologist presented the case of a 42 y.o. male with histologically proven grade III glioma (partial resection). An MRI obtained 3 months after RT-TMZ seemed to demonstrate an increased lesion size. Pre-MTB MRI diagnosis was recurrence, and the treatment hypothesis was a new type of surgery. The neuroradiologist of MTB took into account the following: the patient clinical status (asymptomatic), the MGMT promoter methylation of glioma, the time elapsed after the end of radiotherapy, the contrast enhancement pattern of the lesion (“swiss cheese”), and the ADC values (absolute value at MRI follow-up and compared to the pre-treatment MRI value, both recalculated by the MTB neuroradiologist), and proposed pseudo-progression as the diagnostic hypothesis. Changes in the image interpretation caused changes in the patient management, and the MRI follow-up was proposed without a change in therapy. Six months after the completion of RT, the MRI showed a decreased contrast enhancement and perilesional edema, supporting the hypothesis of pseudo-progression rather than the progression of disease.

The accurate interpretation of the result, of 30.9%, of the discordance between image interpretation performed by the neuroradiologists of our MTB and those by the initial report, requires several considerations. Firstly, these data are similar to those reported by Chung R. et al., albeit in a different area of radiology, and both studies emphasize the role of an expert radiologist as a valued member of the MTB [5]. In the electronic records of the patients referred to our MTB with the MRI performed out of FPG, there was no mention regarding the radiologist who read the MRI and wrote the report. Regardless, based on the MRI protocols and reports, we are quite confident in affirming that in most cases they were general radiologists, or in any case without adequate training in oncological neuroradiology.

Secondly, for the best imaging interpretation in an nMTB, the neuroradiologist should be constantly updated on the therapies used, the therapy chronology, the MRI modifications that the therapies imply, and the potential of all MRI sequences or other imaging techniques to differentiate the treatment changes or disease progression or recurrence. Immunotherapy and anti-angiogenic agents are part of the current standard of treatments and should be taken into account, and there is a tendency to combine therapies (surgery, chemotherapy, radiotherapy, and immunotherapy). The knowledge of post-treatment modifications often comes from the literature, but also from personal experience, as in our center with the first cases of the patients treated with regorafenib [24].

Lastly, cases referred to the nMTB were often complex follow-up cases after multiple treatments, and cases with MRI examinations from different sites. It must therefore be considered that the evaluation of the cases for MTB was carried out in the best conditions for the correct interpretation of the MRI findings, that is with all the previous MRI exams available, to detect even small changes over time or to consider how the pathology appeared before the treatment, and with the chronological history as well as the type of treatment.

From our data, the evaluation of a case by the nMTB required approximately 9 min, a time that tends to increase with an increase in the number of MRI exams to be re-evaluated (median 3 min, min–max 1–12 min).

Obviously, the in-depth evaluation of cases, with the integration of the clinical-anamnestic data, the reassessment of all the previous exams, and sometimes the re-elaboration of diffusion and/or perfusion sequences (guided by new clinical or radiological suspicion), has a cost in terms of time for the neuroradiologist.

From our data, the ODG is one of the histotypes associated with the largest number of cases re-evaluated (12 MRI examinations reviewed for a patient), probably due to the longer life expectancy compared to GB and, therefore, a greater number of MRI exams performed during the follow-up.

The time needed to re-evaluate a case, with the same number of MRI exams, tended to be greater with more externally performed exams. Our explanation is that, in Italy, there is a lack of standardization for the MRI follow-up of brain tumors, not only in terms of the type of sequences, but also the image quality and the modality/timing of the use of contrast medium. It becomes very difficult to compare MRI exams to the sequences acquired on different planes, with different spatial resolution and thicknesses, and the diversity of T1 sequences after contrast media often makes the comparison challenging. Similar differences were reported by S. C. Thust et al. for glioma imaging protocols across Europe [25]. We also realized that no shared guidelines were followed for measuring brain lesions in external sites, bidirectional measurements were not always applied, and volumetric measurements were used in few cases. Therefore, our re-evaluations often deviate from the evaluations made outside because we re-measured the lesions by a standard method in all examinations, obtaining a greater diagnostic accuracy.

In our nMTB, the two principal neuroradiologists (with 30 years of dedicated experience in neuroradiology and 15 years’ experience each) were both present at almost all meetings and applied a double reading in certain difficult cases. The regular participation of this small number of expert neuroradiologists in the nMTB strengthened the rapport with the multidisciplinary team, but the burden of the high volumes of MRI exams and time and staffing constraints have been somewhat burdensome. The burden in terms of the preparation time of our TB for the neuroradiologist is higher than that reported by the national survey published by Snyder et al. [19]. If we consider an average of about 9 min spent for the preparation of a case, and an average number of about 10 cases per MTB session, the preparation effort for the neuroradiologist is about an hour and a half, to which another 60–90 min should be then added, as the duration of the MTB.

This study has several limitations. First, our study was retrospective at a single large tertiary care academic medical center and the sample of cases included might not be representative of other nMTBs in general. It would be useful to compare our data to other national and international nMTBs. The patient cancer outcomes were not captured in this study and can be an area for future research.

## 5. Conclusions

Our study supports the multidisciplinary approach to patient care requiring the commitment of highly specialized physicians and that it can be particularly effective in managing brain tumors. The expert radiologist review impacted patient management in the context of the nMTB, but this has a cost in terms of the time and effort spent preparing for it.

## Figures and Tables

**Figure 1 jpm-12-00135-f001:**
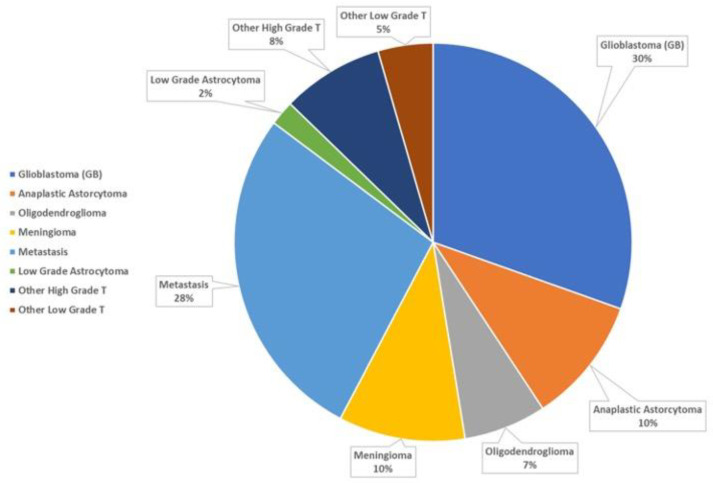
The frequency of the different histotypes of brain tumors discussed over three years at our nMTB.

**Figure 2 jpm-12-00135-f002:**
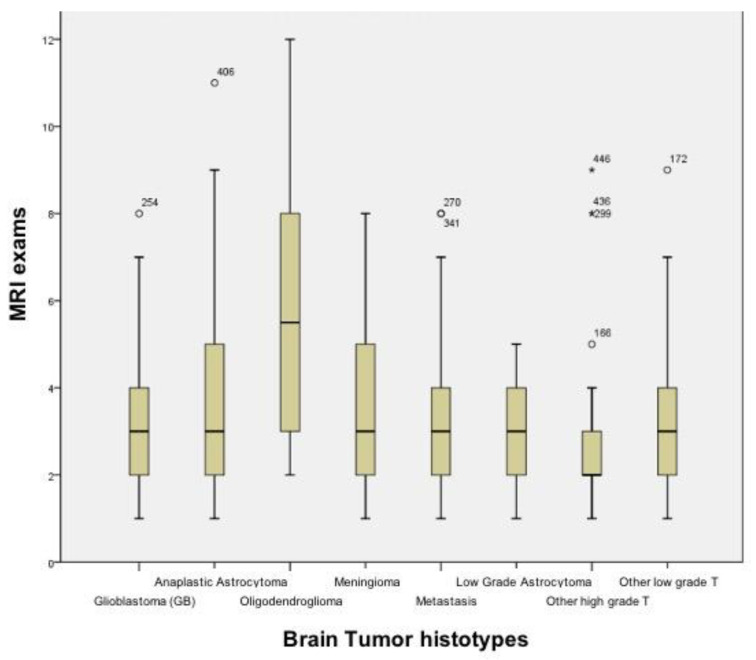
Box plots showing the distribution of the number of MRI exams revised at our nMTB, stratified by brain tumor histotype.

**Figure 3 jpm-12-00135-f003:**
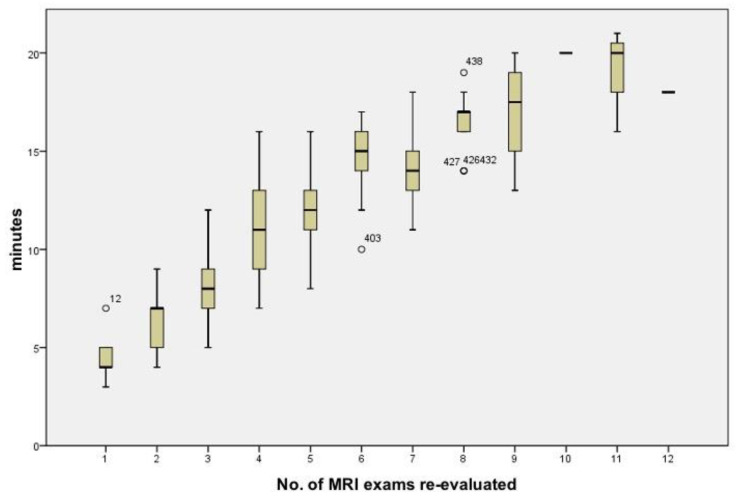
Box plot diagram of the time (minutes) vs. the number of MRI exams evaluated/case.

**Figure 4 jpm-12-00135-f004:**
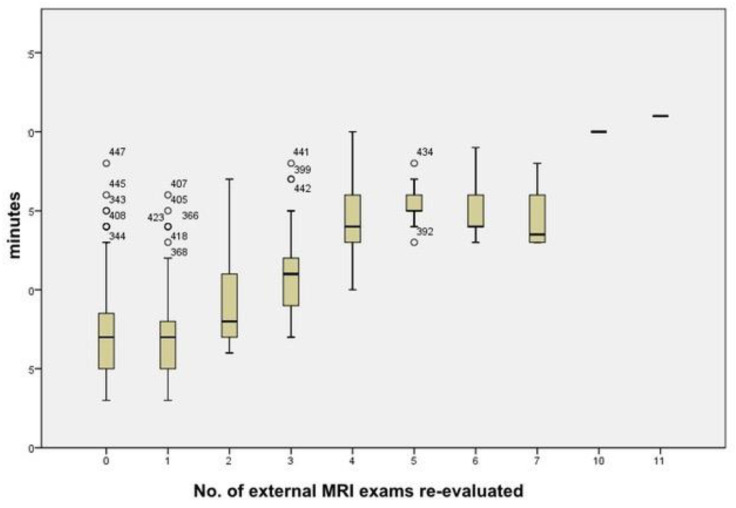
Box plot diagram of the time (minutes) vs. the number of external MRI exams re-evaluated.

**Table 1 jpm-12-00135-t001:** Summary table of the role of the physicians attending our nMTB, divided into who proposed the case (requesting) and to whom questions were addressed (recipient).

Physician	Requesting	Recipient
Radiation oncologist	263	38
Neurosurgeon	181	50
Pathologists	2	29
Neuroradiologist	1	330

**Table 2 jpm-12-00135-t002:** Summary of the consensus recommendations after nMTB.

Consensus Recommendation	Total (n = 447)	Percent
Imaging follow-up	176	39.4
Surgery	65	14.5
Radiation treatment	84	18.8
Chemotherapy	52	11.6
Multimodal treatment	19	4.3
Palliative care	29	6.5
Molecular studies	16	3.6
Nuclear medicine	6	1.3

**Table 3 jpm-12-00135-t003:** Summary of the number of MRI examinations for the patient and their percentage.

Number of MRI Exams/Patient (n = 1514)	Total No. of Patients (n = 447)	Percent
1	58	13
2	121	27.1
3	103	23
4	60	13.4
5	41	9.2
6	29	6.5
7	13	2.9
8	13	2.9
9	4	0.9
10	1	0.2
11	3	0.7
12	1	0.2

**Table 4 jpm-12-00135-t004:** Summary of the evaluation times for each tumor type.

Tumor Types	Minutes Average	SD	Min	Max
GB	9.5	3.6	3	17
AA	9.9	4.4	4	20
ODG	12.7	5.0	4	21
Meningioma	7.8	3.2	3	14
Metastasis	9.0	3.7	3	18
Low grade Astro.	9.1	3.8	4	17
Other high grade T	7.4	2.8	4	16
Other low grade T	10.7	2.6	7	15

## Data Availability

All data are available and provided on request.

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
