# Peer review of "Neuro-Oncology Multidisciplinary Tumor Board: The Point of View of the Neuroradiologist"

_jpm, 2022, doi:10.3390/jpm12020135_

Round 1

Reviewer 1 Report

The authors retrospectively analyzed the 342 cases (447 discussions) discussed in their neuro-oncology multidisciplinary tumor boards (nMTBs) between March/217 and March/2020. They collected the following data:

  • patient data (age, gender tumor type)
  • proposing physician, questions and answers in nMTB
  • changes in patient treatment (CPT)
  • the number of MRI exams, and the time neuro-radiologists’ spent for preparing for the nMTBs.

They found that the neuro-radiologist received many questions from the participant (330/447, 74%) and they needed to spend additionally 9 minutes in average to evaluate the MRIs per patient for nMTB.

They concluded the neuro-radiologists can help patients’ management through nMTBs but they need to spend time for preparation.

The concept of this manuscript is good but I have some concerns.

  1. The benefits to invite neuro-radiologists to nMTBs are not adequately evaluated.

    The authors didn’t define clearly about the CPT. Did the neurosurgeon or the tumor oncologist clearly stated their initial opinion about the patients’ treatment option before the conference? In how many cases the comments from neuro-radiologists actually helped to change the neurosurgeons’ or the tumor oncologists’ mind? If the comments through nMTBs were saved as documents, is it realistic for the authors to go through the documents to find in which cases did the suggestion from neuro-radiologists help CPT. Or, alternatively, the questionnaire for the neurosurgeons or tumor oncologists to ask if they think the suggestion from neuro-oncologists actually helped their CPT might be better to add.

  2. I recommend describing a typical case.

There should be a typical case presentation for the readers’ easy understanding of the discussions in nMTBs. The case who had their treatment changed because of the suggestion from neuro-oncologist in nMTBs should be presented.

Author Response

We thank the reviewer for suggesting improvements to our paper. Based on his first request we have specified better how the MTB was carried out in Material and methods. 2.1 Clinical data: 

“Case presentations of MTB includes patient’s medical history, clinical presentation, diagnostic studies and a “pre-conference” treatment plan. Questions are then asked to one or more specialists to decide on the best possible treatment plan for the patient, after case discussion if the final decision does not correspond to the one proposed at the beginning, it is considered as a changes in patient treatment (CPT).  Interventions of physicians and the final decision concerning each individual patient is formally transcribed and recorded in the MTB report”

We also responded to the first and second requests in the discussions as follows:

“Through MTB reports we traced the final treatment decision and whether or not it corresponded to the a “pre-MTB” treatment plan. In a total of 76.8% CPT, 40.5% it was based on neuroradiologic assessment. A practical example is therefore reported: radiation oncologist presented the case of a 42 y.o. male with histologically proven grade III glioma (partial resection). MRI obtained 3 month after RT-TMZ seemed to demonstrate increased lesion size. Pre-MTB MRI diagnosis was recurrence, and  treatment hypothesis was a new surgery. The neuroradiologist of MTB took into account: the patient clinical status (asymptomatic), the MGMT promoter methylation of glioma, the time elapsed after the end of radiotherapy, the contrast enhancement pattern of the lesion (“swiss cheese”),  and ADC values (absolute value at MRI follow-up and compared to pre-treatment MRI value, both recalculated by MTB neuroradiologist) and proposed pseudo-progression as diagnostic hypothesis. Changes in image interpretation caused changes in patient management, and MRI follow-up was proposed without a change in therapy. Six month after completion of RT MRI showed decreased contrast enhancement and perilesional edema, supporting  the hypothesis of pseudo-progression rather than progression of disease.”

Reviewer 2 Report

The authors summarized their experience with their neuro-oncology multidisciplinary tumor board (nMTB); the data were retrospectively collected from a 10-year period, from a tertiary Italian center. The topic is of interest, as detailed data on nMTBs are rare in the literature. The findings are presented clearly. A few details would need to be further clarified, discussed, as listed below.

Introduction

The authors note, appropriately, that data on nMTBs in the literature are scarce. However, there are far more publications than they list in the manuscript, spanning from 2005 to 2021, and they would be worth mentioning, citing, e.g.:

(i) Lutterbach J, et al. The brain tumour board: lessons to be learned from an interdisciplinary conference. Oncol Res Treat. 2005;28:22-6.

(ii) Snyder J, Schultz L, Walbert T. The role of tumour board conferences in neuro-oncology: a nationwide provider survey. J. Neurooncol. 2017;133:1-7.

(iii) Schäfer N, et al. Implementation, relevance, and virtual adaptation of neuro-oncological tumor boards during the COVID-19 pandemic: a nationwide provider survey. J Neurooncol. 2021 Jul;153(3):479-485. 

Data from these could be also cited in the Discussion (page 5, line 169, where only 1 reference is listed currently) and compared to the present results, in the Discussion.

The authors define the abbreviation MTB, but then they use “MDT” on page 2, line 69. Please use MTB consistently.

Line 90: “revaluation” should be “re-evaluation”; “iconography” – please use “images” instead.

Materials and Methods

The MTBs were attended by “radiation oncologists”, among others; were there any neurologists or neuro-oncologists or general (medical) oncologists participating (none listed). If not, who managed the patients after radiotherapy?

Line 108: please define the abbreviation “FPG”

Statistical analysis: “Student t-test ..for paired samples”; Student t would be used for comparing unpaired groups (and did the data set contain any paired sample for statistics?); also, chi-square test is mentioned, but it is not clear in the Results where this test was used, please clarify.

Results

“Image reinterpretation was discordant with the initial report in 30.9% of cases..” – The authors should clarify who made the original readings vs. the re-interpretations? Were these always different radiologists? This is a critical point, as discordant findings can be expected if independent radiologists read the scans but they are less likely (although possible) if the same radiologist re-reviews the scans (and, with additional clinical information, the opinion may change). – See also a related point in the Discussion below.

“A total of 1514 MRI examinations for 447 patients were evaluated, with a mean number of 3.38 ± SD 1.99 (min-max: 1-12) MRI examinations per patient” – Since the individual numbers were whole numbers, the mean and SD should not be given with 1/100 accuracy; they should be given as 3.4 ± SD 2.0.

“The time employed to review all MRI exams was 67 hours overall, with a mean of 8.98 min for cases…” – The Methods should describe how this time was measured; since this was a retrospective study, how did the authors ascertain the time of each review, accurately retrospectively? Also, if whole minutes were counted, the mean should not be given with 1/100 accuracy; rather, it should be 9.0 min.

“We found a significant difference between GB, meningioma, and ODG concerning evaluation time per case (p<0.05).” – Please give details of these data, i.e., the evaluation times for each tumor type, so that the readers can see which had the longest vs. shortest, etc.

Discussion

Page 6, line 217: “The time needed to re-evaluate a case, with the same number of MRI exams, tended to be greater with more externally performed exams.” – The Results contain no data or any statistical analysis to support this claim.

“The time needed to re-evaluate a case, with the same number of MRI exams, tended to be greater with more externally performed exams.” – The Results should provide data to substantiate this statement.

Author Response

Introduction.  The authors note, appropriately, that data on nMTBs in the literature are scarce. However, there are far more publications than they list in the manuscript, spanning from 2005 to 2021, and they would be worth mentioning, citing, e.g.: (i) Lutterbach J, et al. The brain tumour board: lessons to be learned from an interdisciplinary conference. Oncol Res Treat. 2005;28:22-6. (ii) Snyder J, Schultz L, Walbert T. The role of tumour board  Conferences in neuro-oncology: a nationwide provider survey. J. Neurooncol. 2017;133:1-7. (iii) Schäfer N, et al. Implementation, relevance, and virtual adaptation of neuro-oncological tumor boards during the COVID-19 pandemic: a nationwide provider survey. J Neurooncol. 2021 Jul;153(3):479-485. Data from these could be also cited in the Discussion (page 5, line 169, where only 1 reference is listed currently) and compared to the present results, in the Discussion.

  • We thank the reviewer for the requested insights. We have updated the bibliography with the requested articles, recalled both in the introduction and in the discussions. For a better discussion of Snyder J et al. article we introduced our average number of cases per session into the results. To satisfy the requests of other reviewer as well, we entered comments at different points of the discussion. Below the main changes:
  • Results. “The average number of cases discussed for session ranged from 6 to 15 (mean 10,5), with a total of 447 discussions analyzed. 4. Discussion. As reported by several authors, in both MTB of neuro-oncology or of others sectors, oncologists presented most of the cases (58.8%). The burden in terms of preparation time of our TB for the neuroradiologist is higher than that reported by the national survey published by Snyder et al.. If we consider an average of about 9 minutes spent for the preparation of a case, and an average number of about 10 cases per MTB session, the preparation effort for the neuroradiologist is about an hour and a half, to which should be then added other 60-90 minutes, as the duration of the MTB.”

The authors define the abbreviation MTB, but then they use “MDT” on page 2, line 69. Please use MTB consistently.

  • Done

Line 90: “revaluation” should be “re-evaluation”; “iconography” – please use “images” instead.

  • Done

Materials and Methods The MTBs were attended by “radiation oncologists”, among others; were there any neurologists or neuro-oncologists or general (medical) oncologists participating (none listed). If not, who managed the patients after radiotherapy?  

  • Oncologists also participated at our MTB, we have already indicated their presence in: Material and Methods 2.1 Clinical data, as below:

“At our institution, the nMTB is a weekly meeting, attended by radiation oncologists, oncologists, neurosurgeons, neuropathologists and neuroradiologists.”

Line 108: please define the abbreviation “FPG”.   

  • On reviewer's indication we introduced the explanation for the abbreviation FPG in the Introduction as follows:

“The aim of the present study was to describe the neuro-oncology MTB (nMTB) at our hospital Fondazione Policlinico Universitario Agostino Gemelli IRCCS (FPG)”

Statistical analysis: “Student t-test ..for paired samples”; Student t would be used for comparing unpaired groups (and did the data set contain any paired sample for statistics?); also, chi-square test is mentioned, but it is not clear in the Results where this test was used, please clarify.

  • In our article we have used a paired samples t-test as a dependent sample t-test like a statistical procedure used to determine whether the mean difference between two sets of observations is zero, in particular in the evaluation of patients who underwent to MTB multiple times. In effect the mentioned chi-square test was used during preliminary statistical evaluation of the data and result not reported in manuscript. We have removed the incorrect reference in text.

Results “Image reinterpretation was discordant with the initial report in 30.9% of cases..” – The authors should clarify who made the original readings vs. the re-interpretations?  Were these always different radiologists?  This is a critical point, as discordant findings can be expected if independent radiologists read the scans but they are less likely (although possible) if the same radiologist re-reviews the scans (and, with additional clinical information, the opinion may change). – See also a related point in the Discussion below.

  • Thanks to the reviewer for the questions. To answer fully, we have integrated the answers to the two questions in the text of the discussions in several points as below, also integrating the answers to the requests of other reviewers.

“The accurate interpretation of the result of 30.9% of discordance between image interpretation done by neuroradiologists of our MTB and those by initial report, requires several considerations. Firstly, these data are similar to those reported by Chung R. et al., albeit in a different area of radiology, and both studies emphasize the role of an expert radiologist as a valued member of the MTB [21]. In electronic records of patients referred to our MTB with MRI performed out of FPG, there was no mention regarding the radiologist who read MRI and wrote the report, anyway basing on MRI protocols and reports, we are quite confident in affirming that in most cases they were general radiologists, or in any case without adequate training in oncological neuroradiology. Secondly, for the best imaging interpretation in a nMTB, the neuroradiologist should be constantly updated on the therapies used, the therapy chronology, the MRI modifications that the therapies imply, and the potential of all MRI sequences or other imaging techniques to differentiate treatment changes or disease progression or recurrence. Immunotherapy and anti-angiogenic agents are part of the current standard of treatments and should be taken into account, and there is a tendency to combine therapies (surgery, chemotherapy, radiotherapy, immunotherapy). Knowledge of post-treatment modifications often comes from the literature, but also from personal experience, as in our center with the first cases of patients treated with regorafenib [22]. Lastly, cases referred to the nMTB were often complex follow-up cases after multiple treatments, and cases with MRI examinations from different sites. It must therefore be considered that the evaluation of cases for MTB was carried out in the best conditions for the correct interpretation of the MRI findings, that is with all the previous MRI exams available, to detect even small changes over time or to consider how the pathology appeared before treatment, and with the chronological history  as well as the type of treatment.”

“A total of 1514 MRI examinations for 447 patients were evaluated, with a mean number of 3.38 ± SD 1.99 (min-max: 1-12) MRI examinations per patient” – Since the individual numbers were whole numbers, the mean and SD should not be given with 1/100 accuracy; they should be given as 3.4 ± SD 2.0.

  • Done

“The time employed to review all MRI exams was 67 hours overall, with a mean of 8.98 min for cases…” – The Methods should describe how this time was measured; since this was a retrospective study, how did the authors ascertain the time of each review, accurately retrospectively?

  • When the nMTB of FPG was established, the health management of FPG requested to tabulate the time engagement of radiologists in the board, and that measurement has been reported in the radiological record of MTB. In materials and methods we have specified that we have taken these values from those records

Also, if whole minutes were counted, the mean should not be given with 1/100 accuracy; rather, it should be 9.0 min.

  • Done

“We found a significant difference between GB, meningioma, and ODG concerning evaluation time per case (p<0.05).” – Please give details of these data, i.e., the evaluation times for each tumor type, so that the readers can see which had the longest vs. shortest, etc.

  • We insert table 4, but we have not included the following graph for reasons of space, having already inserted several graphs and tables to comply with the rules of the journal.

Discussion Page 6, line 217: “The time needed to re-evaluate a case, with the same number of MRI exams, tended to be greater with more externally performed exams.” – The Results contain no data or any statistical analysis to support this claim.

  • We have inserted the following paragraph, if the reviewer deems it necessary we can insert the following blox-plot. For reasons of space it had not been inserted, to limit the number of figures to what is indicated by the journal

We calculated the time needed to re-evaluate imaging exams in relation to the variable "number of external MRI", both for the overall exams and for the sub-categories number of exams: e.g. one MR exam, two, three, 4, 5, 6, 7, 8, 9 10,11 and 12 (figure 4). In each subgroup were demonstrated a statistical difference between the group of examination with increasing of time need to re-evaluation with increase number of external exams.

Reviewer 3 Report

To Authors and Editors

"Neuro-oncology multidisciplinary tumor board: the point 2
of view of the neuroradiologist" is quite interesting. Nonetheless, there are some weak points that should be addressed.

1) Please introduce continuous variables as median and interquartile range. Please use Spearman instead of Pearson.

2) Figure 3 is distorted. Please revise

3) Please make the table, using Kruskall Wallis to compare clearly the difference among histotypes and MRI scanning times. I saw that you only introduced BOX-PLOT

4) You mentioned about the correlation test such as SPEARMAN. However, I did not see it in results and discussion? Please clarify.

5) You mentioned about the correlation test such as ANOVA. However, I did not see it in results and discussion? Please clarify.

6) The paper is very simple with n and percentage. You should use more test to compare and find more new results and discuss about them. It will make this paper more attractive.

Sincerely

Author Response

Please introduce continuous variables as median and interquartile range. Please use Spearman instead of Pearson.

You mentioned about the correlation test such as SPEARMAN. However, I did not see it in results and discussion? Please clarify.

  • The preliminary examination of data demonstrate the normal distribution of examinate variable with Kolmogorov‐Smirnov test and Shapiro‐Wilk test (as reported in Statistical Analysis paragraph). Successive statistical evaluation were conducted based on parametric test (Pearson, ANOVA …), and the descriptive data were reported with mean and Standard Deviation. If requested by the reviewer, we can show additional results and statistical analysis.

Please make the table, using Kruskall Wallis to compare clearly the difference among histotypes and MRI scanning times. I saw that you only introduced BOX-PLOT

  • Done

Figure 3 is distorted. Please revise. File.bmp

  • Done

You mentioned about the correlation test such as ANOVA. However, I did not see it in results and discussion? Please clarify.

  • We have used ANOVA in many evaluations, for e.g. in the evaluation of the mean of MRI exams exanimated VS the Histological type of tumor; we have presented only the box-blot in Fig.2. If requested by the reviewer, we can show additional table and statistical analysis. which, however, we believe would lengthen the article too much.

The paper is very simple with n and percentage. You should use more test to compare and find more new results and discuss about them. It will make this paper more attractive.

  • The statistical evaluation of our work have request many statistical test, such as example a Bonferroni correction of ANOVA evaluation of mean of MRI exams exanimated VS the Histological type of tumor, as reported in example, but without any success. We have presented only the most useful data and their graphical representation.

Reviewer 4 Report

Study conducted validly.

Introduction described correctly.

Appropriate materials and methods used.

Statistical analysis conducted according to static models corrected in relation to the variables and variations taken into consideration.

The main innovative field of scientific research conducted is represented by the quantitative and qualitative evaluation of the contribution of each specialist medical doctor included in the multi-disciplinary team, demonstrating the important role played by the often misunderstood neuroradiologist: “review by an expert neuroradiologist impacts patient management in the context of nMTBs but has costs in terms of time and effort spent preparing for it”.

Author Response

We thank the reviewer for the positive comments on our work

Round 2

Reviewer 1 Report

The authors revised all the points that I mentioned and I am totally satisfied with the current revised manuscript.

Author Response

We thank the reviewer for the comments

Reviewer 2 Report

The authors were responsive to my suggestions and made several appropriate adjustments. There are a few remaining issues to address, as follows:

  1. “Continuous variable were evaluated..” – should be “Continuous variables were evaluated…”
  2. “Student’s t test was applied for continuous variables analysis in paired Pearson/Spearman correlations were used to assess relationships between continuous variables.” – This sentence should be rephrased. The authors state in their response that paired-samples t-test was used, and this is not reflected in this sentence which is still confusing.
  3. In the Results, as requested, the authors added the evaluation times for different tumor types. Since the individual numbers for evaluation time (minutes) were whole numbers, the group mean and SD values should be given with no more than 0.1 min accuracy (so, e.g., SD 3,619 should be 3.6).
  4. At the end of the Results, per the reviewer’s suggestion, the authors added the following section:

“We calculated the time needed to re-evaluate imaging exams in relation to the variable "number of external MRI", both for the overall exams and for the sub-categories number of exams: e.g. one MR exam, two, three, 4, 5, 6, 7, 8, 9 10,11 and 12. In each subgroup were demonstrated a statistical difference between the group of examination with increasing of time need to re-evaluation with increase number of external exams.”

I suggest rephrasing this new section as follows: “We calculated the time needed to re-evaluate imaging exams in relation to the variable "number of external MRIs", both for the overall number of exams and for each number of MRIs, from 1-12 exams. The time needed to re-evaluate the MRIs increased with the number of external exams.”

A figure to illustrate this would be helpful and could also give an idea if this is a linear or non-linear increase in time.

Author Response

The authors were responsive to my suggestions and made several appropriate adjustments. There are a few remaining issues to address, as follows:

  1. “Continuous variable were evaluated..” – should be “Continuous variables were evaluated…”

Done

  1. “Student’s t test was applied for continuous variables analysis in paired Pearson/Spearman correlations were used to assess relationships between continuous variables.” – This sentence should be rephrased. The authors state in their response that paired-samples t-test was used, and this is not reflected in this sentence which is still confusing.

We rephrased as follow:

Student’s t test was applied for continuous variables analysis in paired samples, and Pearson/Spearman correlations were used to assess relationships between continuous variables.

  1. In the Results, as requested, the authors added the evaluation times for different tumor types. Since the individual numbers for evaluation time (minutes) were whole numbers, the group mean and SD values should be given with no more than 0.1 min accuracy (so, e.g., SD 3,619 should be 3.6).

Done

  1. At the end of the Results, per the reviewer’s suggestion, the authors added the following section: “We calculated the time needed to re-evaluate imaging exams in relation to the variable "number of external MRI", both for the overall exams and for the sub-categories number of exams: e.g. one MR exam, two, three, 4, 5, 6, 7, 8, 9 10,11 and 12. In each subgroup were demonstrated a statistical difference between the group of examination with increasing of time need to re-evaluation with increase number of external exams.” I suggest rephrasing this new section as follows: “We calculated the time needed to re-evaluate imaging exams in relation to the variable "number of external MRIs", both for the overall number of exams and for each number of MRIs, from 1-12 exams. The time needed to re-evaluate the MRIs increased with the number of external exams.”

We thank the reviewer for the suggested improvements to the text, which we have inserted as indicated

A figure to illustrate this would be helpful and could also give an idea if this is a linear or non-linear increase in time.

Done as follow
